# Sex- and Age-Associated Differences in Genomic Alterations among Patients with Advanced Non-Small Cell Lung Cancer (NSCLC)

**DOI:** 10.3390/cancers16132366

**Published:** 2024-06-27

**Authors:** ErinMarie O. Kimbrough, Julian A. Marin-Acevedo, Leylah M. Drusbosky, Ariana Mooradian, Yujie Zhao, Rami Manochakian, Yanyan Lou

**Affiliations:** 1Division of Hematology and Oncology, Mayo Clinic, Jacksonville, FL 32224, USA; 2Department of Hematology and Oncology, Division of Internal Medicine, Indiana University Melvin and Bren Simon Comprehensive Cancer Center, Indianapolis, IN 46202, USA; 3Guardant Health, Inc., Redwood City, CA 94063, USA; 4Division of Hematology and Medical Oncology, University of Florida, Jacksonville, FL 32209, USA

**Keywords:** non-small cell lung cancer, genomic distribution, mutations, age differences, sex differences

## Abstract

**Simple Summary:**

Lung cancer is the leading cause of cancer-related death. While sex and age impact outcomes in non-small cell lung cancer (NSCLC), it is not clearly understood how these factors affect tumor biology. We believe that sex and age influence the distribution of genomic alterations in NSCLC and evaluated for differences in predictive and/or prognostic alterations in individuals with advanced NSCLC. Our study is the largest, to our knowledge, to evaluate and confirm that the genomic landscape in advanced NSCLC differs by both sex and age.

**Abstract:**

Genomic mutations impact non-small cell lung cancer (NSCLC) biology. The influence of sex and age on the distribution of these alterations is unclear. We analyzed circulating-tumor DNA from individuals with advanced NSCLC from March 2018 to October 2020. *EGFR*, *KRAS*, *ALK*, *ROS1*, *BRAF*, *NTRK*, *ERBB2*, *RET*, *MET*, *PIK3CA*, *STK11*, and *TP53* alterations were assessed. We evaluated the differences by sex and age (<70 and ≥70) using Fisher’s exact test. Of the 34,277 samples, 30,790 (89.83%) had a detectable mutation and 19,923 (58.12%) had an alteration of interest. The median age of the ctDNA positive population was 69 (18–102), 16,756 (54.42%) were female, and 28,835 (93.65%) had adenocarcinoma. Females had more alterations in all the assessed *EGFR* mutations, *KRAS* G12C, and *ERBB2* ex20 ins. Males had higher numbers of *MET* amp and alterations in *STK11* and *TP53*. Patients <70 years were more likely to have alterations in *EGFR* exon 19 del/exon 20 ins/T790M, *KRAS* G12C/D, *ALK*, *ROS1*, *BRAF* V600E, *ERBB2* Ex20ins, *MET* amp, *STK11,* and *TP53*. Individuals ≥70 years were more likely to have alterations in *EGFR* L861Q, *MET* exon 14 skipping, and *PIK3CA*. We provided evidence of sex- and age-associated differences in the distribution of genomic alterations in individuals with advanced NSCLC.

## 1. Introduction

Lung cancer is the third most common cancer in the U.S., after breast and prostate, with an incidence of 50.3 per 100,000 women and 64.1 per 100,000 men [1,2,3]. The median age at diagnosis is 71 years, and non-small cell lung cancer (NSCLC) is uncommon among younger individuals (<50 years) [2,3,4]. Lung cancer is also the leading cause of death among cancer patients. The estimated five-year overall survival is 25.4%, with an estimated death rate of 29.3 per 100,000 women and 42.2 per 100,000 men [1,2,3].

Recent advances, including the discovery of predictive and prognostic biomarkers and the development of targeted therapies, have led to improved outcomes for some patients with lung cancer. Genomic testing, particularly on tumor samples, has become an integral part of the management of NSCLC and is considered a standard of care [5]. A tumor tissue biopsy offers precise histological and molecular information about the tumor, allowing for an accurate diagnosis and a comprehensive review of histology, mutational status, and tumor microenvironment. This approach is limited, however, by the need for invasive procedures and the potential sampling errors due to insufficient tissue or tumor heterogeneity, where a biopsy may fail to capture the complete genetic diversity of the tumor [5]. The concept of a “liquid biopsy” dates back to the 1940s when the molecules released by the primary tumor or metastatic lesions into the peripheral blood were first detected. Modern technologies have allowed us to detect small amounts of tumor in the blood [6]. Next-generation sequencing (NGS) assays are currently utilized to assess circulating-tumor DNA (ctDNA) in clinical practice. ctDNA is particularly helpful when the tumor is difficult to access safely or when repeat biopsies provide inadequate tissue for evaluation and diagnostic testing [6]. ctDNA is more representative of the mutational landscape of the cancer as a whole compared to tissue alone. The testing is less invasive and can be repeated to assess changes in the tumor mutational landscape over time, i.e., real-time monitoring [6]. Limitations of liquid biopsy include its inability to detect some rearrangements/fusions and to distinguish tumor alterations from contaminant leukocyte DNA. Some of the alterations detected may represent clonal hematopoiesis [6]. Detection of rearrangements and fusions using circulating tumor RNA (ctRNA) is thought to be more sensitive than ctDNA. The widespread use of ctRNA has been challenging due to its instability and rapid degradation by ribonucleases and immune cells. ctRNA can be broken down even before it enters circulation [7]. ctRNA may be utilized in conjunction with ctDNA in the future to enhance the detection of targetable alterations [8].

ctDNA is often utilized in NSCLC to detect predictive and/or prognostic mutations that help guide therapeutic decisions [6,9]. The current understanding of these predictive and/or prognostic mutations has improved with more frequent genomic testing and advanced sequencing technologies, but little is known about the factors that influence the distribution of these mutations. Sex and age are thought to impact the genomic landscape and clinical outcomes in patients with NSCLC [4,10,11]. Women with lung cancer have improved median overall survival (OS) compared to men with similar disease and treatment responses, while individuals ≤40 or >70 years tend to have a worse prognosis [4,12,13]. Most of the previously published studies evaluating sex and age differences were conducted using a limited mutation panel and often included a small number of patients [4,10,11].

We conducted a comprehensive analysis to examine how sex and age influence the genomic profiles of advanced NSCLC using one of the largest real-world datasets. We believe that with further characterization of these differences, we can better stratify those with mutations that portend worse outcomes and/or predict response to therapy and improve current treatment strategies.

## 2. Materials and Methods

We conducted a retrospective analysis utilizing de-identified data from the Guardant Health database (Palo Alto, CA, USA) from 1 March 2018 to 26 October 2020. Genomic profiles from patients with advanced NSCLC (stages IIIB and higher), who underwent molecular profiling using the plasma-based ctDNA NGS assay Guardant360, were included for the initial review. Single nucleotide variants (SNV), fusions, indels, and copy number variations (CNV) of up to 83 genes were analyzed. We focused on clinically relevant genomic alterations. Synonymous mutations, variants of undetermined significance (VUS), and sub-clonal mutations were excluded. While we did not have data regarding prior or current treatment, we included only the first serial sample in our analysis to limit the effects that any prior treatment may have on the genomic profiles and to capture newly diagnosed NSCLC patients. To minimize confounding factors that impact the accuracy of liquid biopsy (e.g., low tumor burden), samples without detectable alterations were excluded from the analysis [14,15].

We focused on alterations with known prognostic and/or predictive significance (“alterations of interest”) including *EGFR* exon 19 deletion (ex19 del), *EGFR* exon 20 insertion (ex20 ins), *EGFR* G719X, *EGFR* L858R, *EGFR* T790M, *EGFR* S7681, *EGFR* L861Q, *KRAS G12C*, *KRAS* G12D, *KRAS* G12V, *ALK* fusion, *ROS1* fusion, *BRAF* V600E, *NTRK* 1 fusion, *ERBB2* exon 20 insertion (ex20 ins), *RET* fusion, *MET* exon 14 (ex14) skipping, *MET* amplification (amp) medium, *MET* amp high, any *PIK3CA* mutation, any *STK11* mutation, and any *TP53* mutation. The frequency of these alterations was analyzed according to sex (female and male) and age (<70 and ≥70) using Fisher’s exact test. We selected a cut-off of 70 years of age based on the median age of NSCLC and worse outcomes reported in individuals >70 [4,12]. A *p*-value < 0.05 was considered statistically significant. We also assessed the co-occurrence and mutual exclusivity of these alterations using cBioPortal https://www.cbioportal.org, accessed on 9 January 2022 [16,17].

## 3. Results

Of the 34,277 samples reviewed, 30,790 (89.83%) had a somatic alteration and 19,923 (58.12%) had an alteration of interest (Figure 1). The median age of the ctDNA positive population was 69 (range: 18–102), 16,756 (54.42%) were female, and 28,835 (93.65%) had adenocarcinoma histology (Table 1). *TP53* was the most commonly found mutation of interest (49.08%) followed by *EGFR* (19.92%), *KRAS* (14.39%), *STK11* (7.33%), *MET* (3.36%), *ALK* (1.54%), *PIK3CA* (1.52%), *ERBB2* ex20 ins (1.43%), *BRAF* (1.14%), *ROS1* (0.21%), *RET* (0.07%), and *NTRK1* (0.00%), Appendix A.

### 3.1. Genomic Profile Differences in Advanced NSCLC by Sex

Females with advanced NSCLC had significantly more alterations in all assessed *EGFR* mutations [i.e., ex19 del (9.55% vs. 5.86%; *p* < 0.0001), ex20 ins (1.25% vs. 0.95%; *p* = 0.0139), G719X (0.96% vs. 0.59%; *p* = 0.0003), L858R (6.81% vs. 3.96%; *p* < 0.0001), T790M (1.08% vs. 0.57%; *p* < 0.0001), S768I (0.47% vs. 0.25%; *p* = 0.0013), and L861Q (0.58% vs. 0.34%; *p* = 0.002)], *KRAS* G12C (7.10% vs. 6.41%; *p* = 0.0169), and *ERBB2* ex20 ins (1.40% vs. 1.00%; *p* = 0.0012) compared to the males. Males had significantly higher numbers of *MET* amp medium (1.12% vs. 0.78%; *p* = 0.0024) or high (1.09% vs. 0.60%; *p* < 0.0001), alterations in *STK11* (7.61% vs. 5.10%; *p* < 0.0001) and *TP53* (46.36% vs. 38.02%; *p* < 0.0001) than females. No significant differences between males and females were seen in the frequency of *KRAS* G12D (2.54% vs. 2.46%; *p* = 0.6604) and G12V (2.81% vs. 3.10%; *p* = 0.1383), *ALK* (1.23% vs. 1.38%; *p* = 0.366), *ROS1* (0.21% vs. 0.15%; *p* = 0.274), *BRAF* V600E (1.02% vs. 0.94%; *p* = 0.4848), *NTRK1* (0.00% vs. 0.01%; *p* > 0.9999), *RET* (0.04% vs. 0.08%; *p* = 0.2554), *MET* ex14 skipping (1.09% vs. 1.12%; *p* = 0.827), or *PIK3CA* alterations (1.26% vs. 1.32%; *p* = 0.649), Table 2.

### 3.2. Genomic Profile Differences in Advanced NSCLC by Age

Individuals <70 years of age were more likely to have mutations in *EGFR* ex19 del (10.14% vs. 5.59%; *p* < 0.0001), ex20 ins (1.47% vs. 0.75%; *p* < 0.0001), and T790M (1.04% vs. 0.66%; *p* = 0.0003). They were also more likely to have an alteration of *KRAS* G12C (7.58% vs. 6.01%; *p* < 0.0001) or G12D (2.72% vs. 2.29%; *p* = 0.0176), *ALK* fusion (2.10% vs. 0.51%; *p* < 0.0001), *ROS1* fusion (0.26% vs. 0.09%; *p* = 0.0005), *BRAF* V600E (1.09% vs. 0.86%; *p* = 0.0422), *ERBB2* ex20 ins (1.48% vs. 0.95%; *p* < 0.0001), *MET* amp medium (1.08% vs. 0.79%; *p* = 0.0108) or high (1.10% vs. 0.54%; *p* < 0.0001), *STK11* (7.39% vs. 5.11%; *p* < 0.0001)*,* and *TP53* (44.62% vs. 39.13%; *p* < 0.0001) mutations. Those ≥70 were more likely to have *EGFR* L861Q (0.63% vs. 0.32%; *p* < 0.0001), *MET* ex14 skipping (1.71% vs. 0.52%; *p* < 0.0001)*,* and *PIK3CA* (1.48% vs. 1.12%; *p* = 0.0065) alterations. There were no differences in those <70 versus individuals ≥70 in *EGFR* G719X (0.82% vs. 0.77%; *p* = 0.6528), L858R (5.71% vs. 5.33%; *p* = 0.1475), S768I (0.39% vs. 0.35%; *p* = 0.6405), *KRAS* G12V (3.16% vs. 2.80%; *p* = 0.0649), *NTRK1* (0.01% vs. 0.00%; *p* > 0.9999), or *RET* alterations (0.07% vs. 0.05%; *p* = 0.6477), Table 3.

### 3.3. Sex and Age Differences in Advanced NSCLC

Females <70 years of age had significantly more alterations in *EGFR* ex19 del (12.07% vs. 7.03%; *p* < 0.0001), ex20 ins (1.62% vs. 0.88%; *p* < 0.0001), T790M (1.34% vs. 0.82%; *p* = 0.0013), *KRAS* G12C (8.15% vs. 6.07%; *p* < 0.0001), G12D (2.84% vs. 2.09%; *p* = 0.002), *ALK* (2.23% vs. 0.52%; *p* < 0.0001), *ROS1* (0.21% vs. 0.08%; *p* = 0.0432), *ERBB2* ex20 ins (1.65% vs. 1.16%; *p* = 0.007), *MET* amp high (0.84% vs. 0.35%; *p* < 0.0001), *STK11* (6.33% vs. 3.88%; *p* < 0.0001), and *TP53* (40.51% vs. 35.64%; *p* < 0.0001) than females ≥70. Females ≥70 had significantly higher number of alterations in *EGFR* L861Q (0.77% vs. 0.40%; *p* = 0.0022) and *MET* ex14 skipping (1.69% vs. 0.57%; *p* < 0.0001) compared to females <70. No statistically significant differences were seen in younger versus older females in the frequency of *EGFR* G719X (1.01% vs. 0.91%; *p* = 0.5795), L858R (7.05% vs. 6.60%; *p* = 0.2565), S768I (0.49% vs. 0.46%; *p* = 0.822), *KRAS* G12V (3.34% vs. 2.88%; *p* = 0.0904), *BRAF* V600E (1.07% vs. 0.81%; *p* = 0.0919), *NTRK1* (0.01% vs. 0.00%; *p* > 0.9999), *RET* (0.08% vs. 0.07%; *p* > 0.9999), *MET* amp medium (0.91% vs. 0.65%; *p* = 0.0539), or *PIK3CA* (1.20% vs. 1.46%; *p* = 0.1561) alterations, Table 4.

Males < 70 were more likely to have alterations in *EGFR* ex19 del (7.84% vs. 3.85%; *p* < 0.0001), ex20 ins (1.30% vs. 0.59%; *p* < 0.0001), *KRAS* G12C (6.89% vs. 5.95%; *p* = 0.0229), *ALK* (1.95% vs. 0.49%; *p* < 0.0001), *ROS1* (0.31% vs. 0.10%; *p* = 0.0082), *ERBB2* ex20 ins (1.29% vs. 0.71%; *p* = 0.0006), *MET* amp high (1.41% vs. 0.76%; *p* = 0.0002), *STK11* (8.65% vs. 6.58%; *p* < 0.0001), and *TP53* (49.51% vs. 43.30%; *p* < 0.0001) compared to those ≥70. Males ≥70 had significantly more alterations in *EGFR* L861Q (0.46% vs. 0.23%; *p* = 0.0201), *MET* ex14 skipping (1.73% vs. 0.47%; *p* < 0.0001)*,* and *PIK3CA* (1.50% vs. 1.03%; *p* = 0.0153) compared to those <70. There were no significant differences in alterations between males <70 and ≥70 in *EGFR* G719X (0.59% vs. 0.59%; *p* > 0.9999), L858R (4.12% vs. 3.81%; *p* = 0.3414), T790M (0.27% vs. 0.23%; *p* = 0.7361), S768I (0.68% vs. 0.46%; *p* = 0.0935), *KRAS* G12D (2.57% vs. 2.53%; *p* = 0.8724), G12V (2.94% vs. 2.70%; *p* = 0.4142), *BRAF* V600E (1.12% vs. 0.92%; *p* = 0.2749), *NTRK1* (0.00% vs. 0.00%; *p* > 0.9999), *RET* (0.06% vs. 0.03%; *p* = 0.6875), or *MET* amp medium (1.27% vs. 0.97%; *p* = 0.0921), Table 4.

### 3.4. Co-Occurrence of Mutations of Interest

Most individuals (15,093 or 75.76%) had one alteration of interest; however, 4830 (24.24%) had two or more co-existing mutations of interest (Appendix A). The individual mutation profiles are depicted in the Oncoprint (Appendix A). The heatmap demonstrates co-occurring alterations among those with an alteration of interest (Figure 2). We found *KRAS* and *STK11* alterations often co-occurred (*p* < 0.001). The other alterations tended to occur in isolation (Appendix A).

## 4. Discussion

Outcomes in patients with advanced NSCLC have improved in recent years due to the identification of targetable alterations and the development of novel therapies [2,3,18,19]. While 71% of patients with NSCLC have actionable alterations (e.g., *EGFR*, *ALK*, *BRAF*, *ERBB2*, *MET*, *ROS1*, *RET*, and *KRAS*), some have non-actionable mutations with prognostic and/or predictive implications (e.g., *PIK3CA*, *STK11*, and *TP53*) [20,21,22]. It is important that we understand the factors that influence the mutational landscape of NSCLC because these alterations play a critical role in therapeutic decision-making and affect outcomes. Smaller studies have suggested that sex and age impact the distribution of these alterations [4,10,11]. To our knowledge, our study represents the largest to date to investigate and provide evidence of sex- and age-associated differences in patients with advanced NSCLC. Clinicians can personalize treatment with better understanding of how these differences impact tumor biology.

Our study demonstrates that all evaluated *EGFR* mutations are more common among women, which is consistent with prior reports [4,23,24,25,26,27,28,29]. While some also found that *EGFR* mutations are more common among young individuals, except for *EGFR* L858R, we found that alterations in *EGFR* ex19 del, ex20 ins, and T790M are statistically more common in patients <70, while *EGFR* L861Q is statistically more common in adults ≥70 [4,23,24,25,26,27,28,29]. We found no association with age for mutations in *EGFR* G719X, L858R, or S768I.

There are conflicting data regarding the distribution of *KRAS* mutations in NSCLC. While some studies suggest that *KRAS* is more common in younger women, others have failed to demonstrate these differences [4,28,30,31,32,33,34,35]. In our study, alterations in *KRAS* G12C are more frequent in females and in younger individuals, while *KRAS* G12D mutations are more common in those <70 with no sex differences. We did not find differences in the distribution of *KRAS* G12V by sex or age.

As reported and confirmed in our study, *ALK* fusions are more common among young patients. Data regarding sex differences are mixed, and we found no difference by sex [4,10,36,37,38,39,40,41,42]. *ROS1* fusions are reported more often among younger individuals and females [10,36,43]. In our study, we did not find any significant difference in the distribution by sex, but we confirmed that *ROS1* fusions are significantly more common in younger individuals. Studies have suggested that *BRAF* mutations are more common among females without a clear association with age [4,44,45]. While we found no significant difference by sex in *BRAF* V600E mutations, we did note an association with age <70.

Given the rarity of *NTRK* fusions, sex and age differences are unknown [46,47,48]. We did not observe statistically significant differences by sex or age either, though the sample size was small. *ERBB2* ex20 insertions are reported more frequently among females and younger individuals, and our findings are consistent with these reports [4,49,50]. Studies have failed to demonstrate sex differences in individuals with *RET* fusions, but a trend towards younger age has been reported [51,52]. We found no statistically significant differences by sex or age.

*MET* ex14 skipping mutations are reported more frequently among females and older individuals, while *MET* amplifications are reported more often in males without an association with age [53,54,55]. While we did not find any association with sex, we confirmed that *MET* ex14 skipping mutations are more frequent in individuals ≥70. In addition, we found that *MET* amplifications were associated with male sex and younger age. *PIK3CA* mutations are thought to occur more often in males and older individuals [38,41]. While our results confirm these mutations are more common among men, we did not find any differences by age.

Lastly, our findings are consistent with data suggesting that *STK11* mutations are more common in males and younger individuals and *TP53* mutations are more frequent in males [56,57,58,59,60]. We also found an association with *TP53* mutations and younger age.

While many of our results are consistent with prior studies, several differences exist. Factors such as inclusion of early-stage disease, variability in the genomic testing (e.g., surgical specimens rather than ctDNA), inconsistency in the age cut-offs used to define older and younger populations, and differences in the populations evaluated (U.S. vs. non-U.S.; single institution vs. multi-center) may account for the discrepancies.

Our study highlights the influence sex and age have on the genomic landscape of NSCLC. We found that females were more likely to have alterations with a known targeted therapy (*EGFR*, *KRAS* G12C, *ERBB2* ex20), while males were more likely to have mutations in *STK11* or *TP53* which are associated with poor response to therapy and worse outcomes [61,62]. Similarly, we found that younger patients are more likely to have targetable alterations than older individuals, but their tumors are also more likely to harbor poor prognostic mutations, including *STK11* and *TP53*. These differences may impact the tumor biology, treatment, and outcomes for individuals with NSCLC and help explain why males and individuals ≤40 or >70 tend to have worse prognosis [4,10,11,12]. As we work to improve upon existing treatments, it is important to consider the co-occurrence of these alterations to predict response and/or to develop novel combination strategies. In general, these mutations are thought to be mutually exclusive [63]. While we confirmed that most of the evaluated alterations occur in isolation, we found that *STK11* and *KRAS* were likely to co-occur. This has been described previously and is associated with worse clinical outcomes and poor response to immunotherapy [61,62].

We acknowledge several limitations of our study. Race and tobacco use are known to influence the genomic landscape of NSCLC [64,65]. For example, *EGFR* L858R is three times more common in Asian individuals, while *KRAS* G12C is most common among white individuals [64]. Tobacco use is more common among males and individuals <70) [65]. Tumor mutational burden increases with tobacco use, while driver mutations are more common in never-smokers [66,67]. Unfortunately, we did not have access to this demographic information and could not assess how race and tobacco use influenced the genomic landscape of our study population. In addition, information regarding the specific histology was limited and clinical outcome data were not available. While we attempted to identify patients with newly diagnosed advanced NSCLC and minimize the effect of prior treatment on the genomic profile by using only the first serial sample, we were unable to distinguish the mutational landscape differences between treatment naïve individuals and those who received therapy. We recognize that prior treatment may have influenced the tumor mutational landscape as a mechanism of resistance to therapy. Finally, cross comparison of our study to others in the literature was challenging given the different age cut-offs used to define older and younger adult populations, differences in the tissue testing (plasma, tumor, or other), inclusion of early-stage cancers, and differences in the populations evaluated (e.g., Asian or European vs. North American; single institution vs. multi-center).

## 5. Conclusions

To our knowledge, this is one of the largest studies to date evaluating the genomic landscape differences in individuals with advanced NSCLC by sex and age. We demonstrated significant differences in the distribution of predictive and/or prognostic alterations according to sex and age. This could explain differences in outcomes in otherwise similar patients (i.e., stage, histology, and performance status). Further research is needed to understand how these mutations interact with one another, affect response to therapy, and how they can be used to expand on existing therapies to improve outcomes. The influence that race, tobacco use, and other environmental stressors have on the distribution of these alterations by sex and age should also be evaluated.

## Figures and Tables

**Figure 1 cancers-16-02366-f001:**
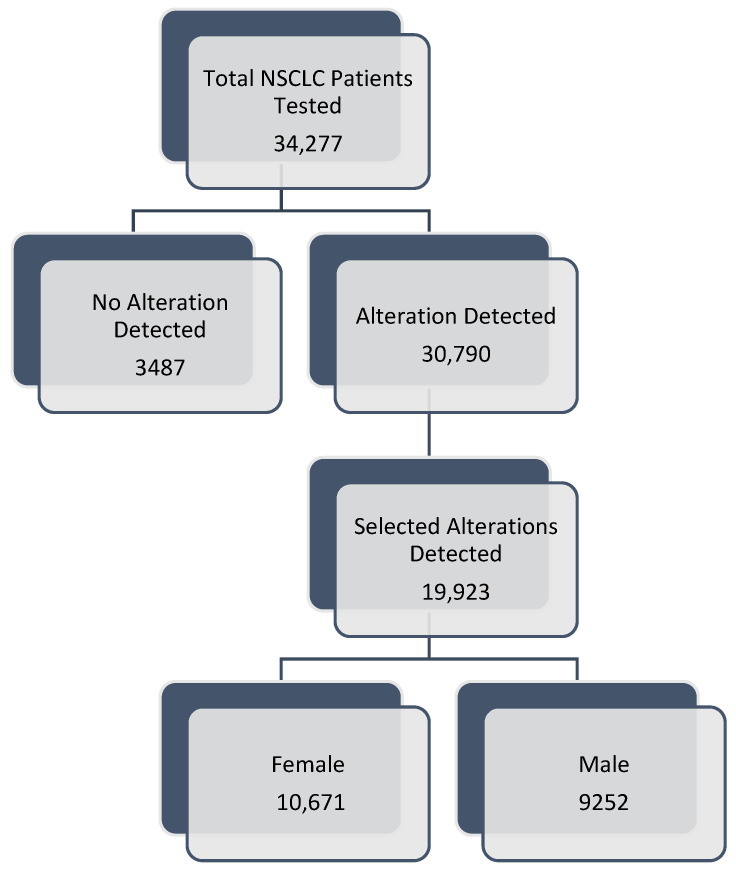
Study Population. The basic study design and population are depicted above. We evaluated the individual profiles of patients with advanced NSCLC who underwent molecular profiling using ctDNA from the Guardant Health database. Individuals without an alteration detected were excluded. Among those with an alteration, we focused on individuals with variants of predictive or prognostic significance including *EGFR* exon 19 deletion, *EGFR* exon 20 insertion, *EGFR* G719X, *EGFR* L858R, *EGFR* T790M, *EGFR* S7681, *EGFR* L861Q, *KRAS* G12C, *KRAS* G12D, *KRAS* G12V, *ALK* fusion, *ROS1* fusion, *BRAF* V600E, *NTRK1* fusion, *ERBB2* exon 20 insertion, *RET* fusion, *MET* exon 14 skipping, *MET* amplification medium, *MET* amplification high, any *PIK3CA* mutation, any *STK11* mutation, or any *TP53* mutation. Of the 34,277 samples reviewed, 30,790 (89.83%) had a somatic alteration and 19,923 (58.12%) had an alteration of interest. Abbreviations: NSCLC, non-small cell lung cancer; ctDNA, circulating tumor DNA.

**Figure 2 cancers-16-02366-f002:**
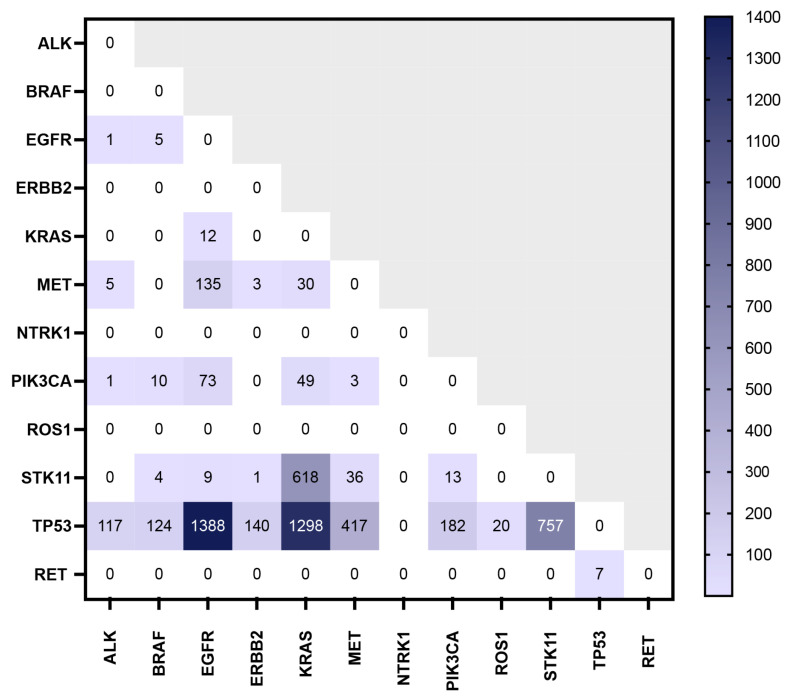
Heatmap of Alterations of Interest. This heatmap shows the co-occurring mutation profiles for patients with an alteration of interest (*n* = 19,923).

**Table 1 cancers-16-02366-t001:** ctDNA-positive patient demographic information.

ctDNA+ Population *n* = 30,790			
Characteristics	<70 Years *n* = 15,495 (50.32%)	≥70 Years *n* = 15,237 (49.49%)	Age Unknown *n* = 58 (0.19%)
Median Age (Range)	61 (18–69)	77 (70–102)	N/A
Sex			
Female	8416 (50.23%)	8307 (49.58%)	33 (0.20%)
Male	7079 (50.44%)	6930 (49.38%)	25 (0.18%)
Histologic Subtype			
Adenocarcinoma	14,530 (50.39%)	14,247 (49.41%)	58 (0.20%)
NSCLC NOS	965 (49.36%)	990 (50.64%)	0 (0.00%)

Basic demographic information for the ctDNA positive study cohort is provided in this table. The median age of the ctDNA positive population was 69 (range 18–102), 16,756 (54.42%) were female, and 28,835 (93.65%) had adenocarcinoma histology. Abbreviations: ctDNA, circulating tumor DNA; N/A, not applicable; NOS, not otherwise specified; NSCLC, non-small cell lung cancer.

**Table 2 cancers-16-02366-t002:** ctDNA+ population with alterations of interest according to sex.

Alteration Detected		Total Patients (*n* = 30,790)		*p*-Value
		Females	Males	
		16,756 (54.42%)	14,034 (45.58%)	
*EGFR*	Exon 19 deletion	1600 (9.55%)	822 (5.86%)	<0.0001
	Exon 20 insertion	209 (1.25%)	133 (0.95%)	0.0139
	G719X	161 (0.96%)	83 (0.59%)	0.0003
	L858R	1141 (6.81%)	556 (3.96%)	<0.0001
	T790M	181 (1.08%)	80 (0.57%)	<0.0001
	S768I	79 (0.47%)	35 (0.25%)	0.0013
	L861Q	98 (0.58%)	48 (0.34%)	0.002
*KRAS*	G12C	1190 (7.10%)	900 (6.41%)	0.0169
	G12D	413 (2.46%)	357 (2.54%)	0.6604
	G12V	520 (3.10%)	395 (2.81%)	0.1383
*ALK*	Fusion	231 (1.38%)	172 (1.23%)	0.366
*ROS1*	Fusion	25 (0.15%)	29 (0.21%)	0.274
*BRAF*	V600E	157 (0.94%)	143 (1.02%)	0.4848
*NTRK*	NTRK 1 Fusion	1 (0.01%)	0 (0.00%)	>0.9999
*ERBB2*	Exon 20 insertions	235 (1.40%)	140 (1.00%)	0.0012
*RET*	Fusion	13 (0.08%)	6 (0.04%)	0.2554
*MET*	Exon 14 skipping	188 (1.12%)	153 (1.09%)	0.827
	Amplification medium	131 (0.78%)	157 (1.12%)	0.0024
	Amplification high	100 (0.60%)	153 (1.09%)	<0.0001
*PIK3CA*	Mutant	222 (1.32%)	177 (1.26%)	0.649
*STK11*	Mutant	855 (5.10%)	1068 (7.61%)	<0.0001
*TP53*	Mutant	6370 (38.02%)	6506 (46.36%)	<0.0001

Differences in the genomic alterations by sex in the ctDNA-positive population are depicted above. A *p*-value < 0.05 indicates statistical significance. Abbreviations: ctDNA, circulating tumor DNA.

**Table 3 cancers-16-02366-t003:** ctDNA+ population with alterations of interest according to age.

Alteration Detected		Total Patients (*n* = 30,732)		*p*-Value
		Patients < 70	Patients ≥ 70	
		15,495 (50.42%)	15,237 (49.58%)	
*EGFR*	Exon 19 deletion	1571 (10.14%)	851 (5.59%)	<0.0001
	Exon 20 insertion	228 (1.47%)	114 (0.75%)	<0.0001
	G719X	127 (0.82%)	117 (0.77%)	0.6528
	L858R	885 (5.71%)	812 (5.33%)	0.1475
	T790M	161 (1.04%)	100 (0.66%)	0.0003
	S768I	60 (0.39%)	54 (0.35%)	0.6405
	L861Q	50 (0.32%)	96 (0.63%)	<0.0001
*KRAS*	G12C	1174 (7.58%)	916 (6.01%)	<0.0001
	G12D	421 (2.72%)	349 (2.29%)	0.0176
	G12V	489 (3.16%)	426 (2.80%)	0.0649
*ALK*	Fusion	326 (2.10%)	77 (0.51%)	<0.0001
*ROS1*	Fusion	40 (0.26%)	14 (0.09%)	0.0005
*BRAF*	V600E	169 (1.09%)	131 (0.86%)	0.0422
*NTRK*	NTRK 1 Fusion	1 (0.01%)	0 (0.00%)	>0.9999
*ERBB2*	Exon 20 insertions	230 (1.48%)	145 (0.95%)	<0.0001
*RET*	Fusion	11 (0.07%)	8 (0.05%)	0.6477
*MET*	Exon 14 skipping	81 (0.52%)	260 (1.71%)	<0.0001
	Amplification medium	167 (1.08%)	121 (0.79%)	0.0108
	Amplification high	171 (1.10%)	82 (0.54%)	<0.0001
*PIK3CA*	Mutant	174 (1.12%)	225 (1.48%)	0.0065
*STK11*	Mutant	1145 (7.39%)	778 (5.11%)	<0.0001
*TP53*	Mutant	6914 (44.62%)	5962 (39.13%)	<0.0001

Differences in the genomic alterations by age in the ctDNA-positive population are depicted above. Only *n* = 30,732 instead of *n* = 30,790 were included in this analysis because of missing age data for 58 patients. A *p*-value < 0.05 indicates statistical significance. Abbreviations: ctDNA, circulating tumor DNA.

**Table 4 cancers-16-02366-t004:** ctDNA+ population with alterations of interest according to sex and age.

Alteration Detected		Total Females (*n* = 16,723)		*p*-Value	Total Males (*n* = 14,009)		*p*-Value
		Females < 70	Females ≥ 70		Males < 70	Males ≥ 70	
		8416 (27.39%)	8307 (27.03%)		7079 (23.03%)	6930 (22.55%)	
*EGFR*	Exon 19 deletion	1016 (12.07%)	584 (7.03%)	<0.0001	555 (7.84%)	267 (3.85%)	<0.0001
	Exon 20 insertion	136 (1.62%)	73 (0.88%)	<0.0001	92 (1.30%)	41 (0.59%)	<0.0001
	G719X	85 (1.01%)	76 (0.91%)	0.5795	42 (0.59%)	41 (0.59%)	>0.9999
	L858R	593 (7.05%)	548 (6.60%)	0.2565	292 (4.12%)	264 (3.81%)	0.3414
	T790M	113 (1.34%)	68 (0.82%)	0.0013	19 (0.27%)	16 (0.23%)	0.7361
	S768I	41 (0.49%)	38 (0.46%)	0.822	48 (0.68%)	32 (0.46%)	0.0935
	L861Q	34 (0.40%)	64 (0.77%)	0.0022	16 (0.23%)	32 (0.46%)	0.0201
*KRAS*	G12C	686 (8.15%)	504 (6.07%)	<0.0001	488 (6.89%)	412 (5.95%)	0.0229
	G12D	239 (2.84%)	174 (2.09%)	0.002	182 (2.57%)	175 (2.53%)	0.8724
	G12V	281 (3.34%)	239 (2.88%)	0.0904	208 (2.94%)	187 (2.70%)	0.4142
*ALK*	Fusion	188 (2.23%)	43 (0.52%)	<0.0001	138 (1.95%)	34 (0.49%)	<0.0001
*ROS1*	Fusion	18 (0.21%)	7 (0.08%)	0.0432	22 (0.31%)	7 (0.10%)	0.0082
*BRAF*	V600E	90 (1.07%)	67 (0.81%)	0.0919	79 (1.12%)	64 (0.92%)	0.2749
*NTRK*	NTRK 1 Fusion	1 (0.01%)	0 (0.00%)	>0.9999	0 (0.00%)	0 (0.00%)	>0.9999
*ERBB2*	Exon 20 insertions	139 (1.65%)	96 (1.16%)	0.007	91 (1.29%)	49 (0.71%)	0.0006
*RET*	Fusion	7 (0.08%)	6 (0.07%)	>0.9999	4 (0.06%)	2 (0.03%)	0.6875
*MET*	Exon 14 skipping	48 (0.57%)	140 (1.69%)	<0.0001	33 (0.47%)	120 (1.73%)	<0.0001
	Amplification medium	77 (0.91%)	54 (0.65%)	0.0539	90 (1.27%)	67 (0.97%)	0.0921
	Amplification high	71 (0.84%)	29 (0.35%)	<0.0001	100 (1.41%)	53 (0.76%)	0.0002
*PIK3CA*	Mutant	101 (1.20%)	121 (1.46%)	0.1561	73 (1.03%)	104 (1.50%)	0.0153
*STK11*	Mutant	533 (6.33%)	322 (3.88%)	<0.0001	612 (8.65%)	456 (6.58%)	<0.0001
*TP53*	Mutant	3409 (40.51%)	2961 (35.64%)	<0.0001	3505 (49.51%)	3001 (43.30%)	<0.0001

Differences in the genomic alterations by sex and age in the ctDNA-positive population (*n* = 30,790) are depicted above. Only *n* = 30,732 were included in this analysis because of missing age data for 58 patients. A *p*-value < 0.05 indicates statistical significance. Abbreviations: ctDNA, circulating tumor DNA.

## Data Availability

The datasets presented in this article are not readily available. The data analyzed in this study were obtained from Guardant Health database. Requests to access these datasets will require a data use agreement and can be requested by contacting medicalaffairs@guardanthealth.com.

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
