# Peer review of "Sex- and Age-Associated Differences in Genomic Alterations among Patients with Advanced Non-Small Cell Lung Cancer (NSCLC)"

_cancers, 2024, doi:10.3390/cancers16132366_

Round 1

Reviewer 1 Report

Comments and Suggestions for Authors

The Authors analyzed circulating-tumor-DNA  from 34,277 individuals with advanced NSCLC  with Guardant 360 assay and reported the prevalence of driver alterations and their distribution according sex and age of the patients.

The report is well written, readable and the huge data number is well shown.

There are some observation that I would like to submit to the Authors.

a)     They in the paragraph “ Material and Methods” report :

“Genomic profiles from patients with advanced NSCLC (stages IIIB and higher) who underwent molecular profiling using the plasma-based ctDNA NGS assay Guardant360 were included for initial review”

In the “Discussion” they write:

“we are unable to distinguish mutational landscape differences between treatment naïve individuals and those who received therapy “

There is an apparent contradiction.

b)     How Guardant 360 can reduce the potential false positive results identifying CHIP mutations?

c)     Was the tumor fraction considered in each evaluated sample?

d)     The performance of the assay varies according to  the type of alteration , the reportable range and  allelic fraction/copy number. How these parameters were taken into consideration in the study?

e)     There is a widespread notion that ctDNA testing is suboptimal for detecting genetic fusions compared with tissue testing.  Recently it was shown that he prevalence of driver rearrangements in tissue and liquid biopsies was comparable when TF ≥1%. ( Kasi PM, Lee JK, Pasquina LW, Decker B, Vanden Borre P, Pavlick DC, Allen JM, Parachoniak C, Quintanilha JCF, Graf RP, Schrock AB, Oxnard GR, Lovly CM, Tukachinsky H, Subbiah V. Circulating Tumor DNA Enables Sensitive Detection of Actionable Gene Fusions and Rearrangements Across Cancer Types. Clin Cancer Res. 2024 Feb 16;30(4):836-848. doi: 10.1158/1078-0432.CCR-23-2693. PMID: 38060240; PMCID: PMC10870120.)

These Authors reported that among the 53,842 examined liquid biopsies, 14% (7,377) contained at least one pathogenic rearrangement. In the present study the incidence of ALK, ROS1 and RET rearrangements report is very low. How they explain these results?

It has been proposed that RNA-based methods of rearrangement detection have superior sensitivity than DNA-based techniques, because they do not require sequencing intronic regions, and profile highly expressed transcript. Do the Authors agree with this statement?

Reviewer 2 Report

Comments and Suggestions for Authors

The authors' study, a significant and comprehensive ctDNA analysis in NSCLC, provides a wealth of data. The authors have appropriately cited the relevant papers and summarized them. I have a few suggestions for revisions.

My comments are listed below.

Major comments:

1.       The figure in Table 1 could be more precise to read visually and in contrast. The authors should reconsider the color choices and review and adjust the placement of the text.

2.       The authors mentioned that they could not distinguish between untreated and previously treated patients. Why is that? If possible, please provide data for untreated cases only.

Reviewer 3 Report

Comments and Suggestions for Authors

In this article, Kimbrough et al. have shown a significant correlation between sex and age with the incidence of genomic alterations in a large cohort of lung cancer patients. This is a significant advance in terms of patient stratification and access to precision therapy. The article is well written and also well structured.

However, the lack of data on tobacco use, race, and response to treatment are the weaknesses of this article. It is important to re-evaluate the effect of genotoxic stress related to tobacco consumption as a function of age and sex. The authors need to further discuss the importance of these key points in understanding these age- and sex-related differences. We need to propose a further study that takes into account the clinical and environmental data of the patients. All of the preclinical studies have been done in male mice, but it is important to take into account this difference between men and women. This will lead to a new specific therapeutic strategy. 

Comments on the Quality of English Language

the article is well written

Round 2

Reviewer 1 Report

Comments and Suggestions for Authors

No more suggestions